# Propagation of *Babesia bigemina* in Rabbit Model and Evaluation of Its Attenuation in Cross-Bred Calves

**DOI:** 10.3390/ani12172287

**Published:** 2022-09-03

**Authors:** Naimat Ullah, Kamran Ashraf, Abdul Rehman, Muhammad Suleman, Muhammad Imran Rashid

**Affiliations:** 1Department of Parasitology, Faculty of Veterinary Science, University of Veterinary and Animal Sciences, Lahore 54000, Pakistan; 2Department of Epidemiology and Public Health, Faculty of Veterinary Science, University of Veterinary and Animal Sciences, Lahore 54000, Pakistan; 3Institute of Microbiology, Faculty of Veterinary Science, University of Veterinary and Animal Sciences, Lahore 54000, Pakistan

**Keywords:** *Babesia bigemina*, in vivo propagation, attenuation, rabbit, calves

## Abstract

**Simple Summary:**

Vector-borne diseases (VBDs) cause heavy economic losses in the livestock sector. Among these VBDs, babesiosis is the second most common disease, causing high morbidity, mortality, and reproductive and productive losses in cattle. The causative agents of this disease are globally distributed across tropical and subtropical countries. In the current study, *B. bigemina* was propagated in vivo in rabbit for attenuation and evaluated for its virulence periodically. This attenuated *B. bigemina* was then inoculated to naive calves for the evaluation of clinical parameters. Increased parasitaemia and temperature were observed in rabbits following the inoculation of *B. bigemina*-infected red blood cells. The naive calves did not show symptoms of parasitaemia or temperature elevation when inoculated with rabbit-propagated *B. bigemina*-infected RBCs. Furthermore, this study also demonstrated that infected cattle periodically had a decreased PCV profile, along with increased temperature and parasitaemia. Moreover, this study also revealed correlations between the temperature, parasitaemia, and packed cell volume of inoculated, infected, and control group calves.

**Abstract:**

Bovine babesiosis (BB) is a vector-borne disease (VBD) that affects cattle in tropical and subtropical areas, caused by the haemoprotozoa *Babesia bovis* and *Babesia bigemina*. It is transmitted by tick bites belonging to the genus *Rhipicephalus* and is clinically characterized by high fever, depression, anorexia, decreased milk and meat production, haemoglobinemia, haemoglobinuria, jaundice, and pregnancy loss. In this study, the propagation of *B. bigemina* was evaluated by intraperitoneally inoculating 3 × 10^6^ red blood cells infected with *B. bigemina* into rabbits. The study showed that variations in rabbit body temperatures are related to induced bovine babesiosis. A significant increase in temperature (39.20 ± 0.23 °C) was observed from day 4 onwards, with the maximum temperature (40.80 ± 1.01 °C) on day 9 post-inoculation. This study included susceptible cross-bred calves for in vivo attenuation, and they were compared with an infected group. The calves in the infected group showed a significant increase in temperature (38.79 ± 0.03 °C) from day 3 onwards and a maximum temperature (41.3 ± 0.17 °C) on day 11. Inoculated calves showed a gradual rise in temperature post-inoculation, but the difference was not significant. Inoculated calves did not show parasitaemia, whereas 32% of infected calves displayed parasitaemia. As compared to inoculated calves post-inoculation, packed cell volume (PCV) decreased (16.36 ± 1.30) for infected calves. However, there were statistically significant differences (*p* ≤ 0.05) in temperatures, parasitaemia, and PCV in both inoculated and infected calves. The current study aimed to attenuate *B. bigemina* in rabbit models and evaluate the pathogenic potential of this organism in naive calves. In conclusion, *B. bigemina* proliferation was attenuated in rabbits. The rabbit model can be used to study *B. bigemina* in vivo in order to reduce its pathogenicity.

## 1. Introduction

Victor Babes first described the parasite that is usually found in the blood of cattle and causes haemoglobinuria, which was later renamed *B. bovis* in his honour [1,2,3]. Similarly, Smith and Kilborne identified a tick-borne haemoparasite (*Boophilus annulatus*) in 1893 responsible for the transmission of Texas Fever to cattle [4]. This parasite was named *B. bigemina* [5,6].

Babesiosis is the leading haemoparasitic disease of animals, causing heavy economic losses in endemic areas [7]. BB is a vector-borne disease of cattle that is prevalent in the tropics and subtropics. It is caused by a haemoprotozoan of the genus *Babesia*. There are two economically important species of *Babesia* that cause Babesiosis, namely, *B. bovis* and *B. bigemina*. It is transmitted by the bite of ticks in the genus *Rhipicephalus*, including *R. microplus*, *R. annulatus,* and *R. decoloratus* [8,9,10]. VBDs are estimated to cost the global economy USD 13.9–18.7 billion per year [11]. BB threatens approximately 1–2 billion cattle worldwide [7]. BB is clinically characterized by fever, depression, anorexia, decreased milk and meat production, haemoglobinemia, haemoglobinuria, jaundice, abortion, and mortality [7,12,13,14]. Usually, it is diagnosed using a microscopic examination of blood smears, clinical examination, and serological tests (IFAT, CFT, and PCR) [9,15,16]. VBD can be controlled with chemoprophylaxis, vector control, and immunization [17].

The artificial control of tick populations has been suggested. Despite the fact that vector control is achieved by using chemoprophylaxis with acaricides, their prolonged use may generate resistance [18]. One of the two most commonly used babesicides, imidocarb at a dose rate of 3 mg per kg, has been found to provide an adequate premunition against *B. bigemina* and *B. bovis* upon the inoculation of a bivalent *Babesia* vaccine after 21 and 61 days, respectively [19]. Imidocarb needs to be eliminated because it has residues in human food and is used as a preventative measure when cattle are exposed to babesiosis during their protective period [20,21,22]. As a result, alternative strategies for safer and more effective control are required.

Attenuation refers to a decrease in the virulence of pathogenic species. There are two different methods that are most commonly used for attenuating pathogenic species of cattle, which are in vivo and in vitro. It has been reported that successive passages of *B. bovis* result in progressively less severe signs of disease and reduced virulence in splenectomized calves. After eight to twenty calf passages, *B. bovis* attenuation was observed [23]. Standard vaccines have been prepared by inoculating attenuated strains of *Babesia* species into splenectomized cattle, followed by the collection of blood [24,25]. A microaerophilic stationary phase (MASP) culture technique has been developed for attenuating *B. bovis* [26]. In the MASP technique, *B. bovis* is cultured continuously in a layer of bovine RBCs in a culture medium under low oxygen pressure. It has been shown that *B. bigemina* can be propagated in vitro through continuous culture [27]. *B. bigemina* was continuously cultured in bovine RBCs with 60% medium 199, 40% bovine serum, pH 7, 10% PCV, and 5% CO_2_ at 37 °C. In a rabbit model, *T. annulata* has been propagated in vivo and attenuated in cross-bred calves [28].

To the best of our knowledge, *B. bigemina* has not yet been propagated in a rabbit model for the purposes of attenuation. Therefore, the aim of the present study was to propagate *B. bigemina* in rabbits and evaluate its attenuation in naive cross-bred calves.

## 2. Materials and Methods

### 2.1. Source of Parasite

Using the methodology adopted earlier [29], with modifications, a local strain of *B. bigemina* was isolated from infected cross-bred cattle, and the infection was sustained in calves. An EDTA-containing vacutainer was used to collect blood aseptically from cattle calves diagnosed with babesiosis (acute stage) exhibiting clinical symptoms (41.1 °C, anaemia, and haemoglobinuria). Blood was collected from restrained cattle calves by puncturing their jugular veins with a sterile syringe and antiseptic gauze. It was transferred to a vacutainer containing EDTA. Blood smears were prepared after Giemsa staining and observed at 100× magnification (oil-immersion lens). These blood samples were subjected to polymerase chain reaction (PCR) to confirm the presence of *B. bigemina*-specific primers.

### 2.2. Inoculation into Rabbit

Six healthy male rabbits (Oryctolagus cuniculus) up to six months old and weighing 1350–1750 g, were purchased from a local market [30]. Each individual was quarantined on its own and given food ad libitum, fresh water, and standard palate feed. A microscopic examination and PCR were used to screen blood samples from the rabbits’ ears for haemoparasites such as *Babesia, Theileria,* and *Anaplasma*. Both microscopy and PCR showed no evidence of infection in these rabbits. The G-STORM Thermocycler (catalog no. GS04822; G-STORM, Somerset, UK) was used to perform PCR following the conditions of [31,32], with slight modifications based on specific primers of the 18srRNA gene (*B. bigemina* F = 5-AGAGGGACTCCTGTGCTTCA-3, *B. bigemina* R = 5-GACGAATCGGAAAAGCCACG-3) in order to obtain a product length of 321 bp for confirmed *B. bigemina*. For the PCR recipe, a final volume of 25 µL was prepared. The denaturation process was carried out at 95 degrees for 1 min, followed by 37 cycles of PCR. Each cycle included denaturation (95 °C for 30 s), annealing (56 °C for 30 s), extension (72 °C for 30 s), and final elongation (72 °C for 5 min). The DNA fragments were analysed on agarose gel with a 1.5% concentration.

A dose of 3 × 10^6^ red blood cells (RBCs) infected with *B. bigemina* from cattle in the acute stage were inoculated intraperitoneally into rabbits simultaneously following the protocol used earlier [33], with slight modifications of parasitaemia estimation at every 4-day interval. The piroplasm of *B. bigemina* was observed in blood samples of rabbits. The body temperatures of rabbits were monitored daily. The blood samples were collected for their microscopy, parasitaemia monitoring, and PCR every 4 days up to 4 weeks [34]. Blood was collected from rabbits and subjected to PCR on day 8 post-inoculation [35].

### 2.3. Inoculation of Parasite into Calves

During the study, (*n* = 15) cross-bred 4–8-month-old male calves (weighing about 60–80 kg) were purchased from a local market and evaluated for any infection using microscopic and PCR analysis. The calves were acclimatized in an animal research station at UVAS, Lahore, by providing ad libitum feed and water. After screening, the calves were categorized into three groups (five calves/group): inoculated (attenuated *B. bigemina*), infected (virulent *B. bigemina*), and control (uninfected).

Parasitized blood was collected from six rabbits by saphenous vein puncture, and a calculated dose of rabbit-attenuated *B. bigemina*-infected RBCs (2 × 10^6^ iRBCs) with 4% PPE were separated and subcutaneously inoculated on the 7th day into the inoculated group. Infected calves were intravenously administered virulent *B. bigemina* (5 × 10^8^ iRBCs) derived from an experimentally infected calf following a previously published protocol [29]. The control group was inoculated with a similar dose of uninfected RBCs from a healthy rabbit. All the groups (inoculated, infected, and control) were monitored daily for temperature, while parasitaemia and packed cell volume (PCV) were measured at intervals of the 4th day and 7th day post-inoculation up to 4 weeks [36].

### 2.4. Statistical Analysis

Based on the results of the infected and inoculated calves, a statistical analysis was performed using the *t*-test and analysis of variance (ANOVA). Data were analysed using GraphPad Prism version 6 software (GraphPad Software 7825 Fay Avenue, Suite 230, La Jolla, CA, USA). An unpaired *t*-test was used to compare differences between independent groups, while a repeated measures ANOVA was used to compare the mean differences between infected and inoculated calves [37].

## 3. Results

### 3.1. The Effect of Babesiosis on Body Temperature and Parasitaemia in Rabbits

The results revealed a correlation between induced Babesiosis and fluctuations in temperature in rabbits with parasitaemia. As early as day 4 post-inoculation (39.20 ± 0.23 °C) up to day 9, there was an increase in body temperature among all experimental rabbits. On the 9th day post-inoculation, a maximum increase in body temperature (40.80 ± 1.01 °C) was observed, whereas on the 10th day post-inoculation, body temperature (38.65 ± 0.10 °C) and parasitaemia gradually decreased. Among all experimental rabbits, parasitaemia developed on the 4th day PI and reached 24% by the 9th day, as indicated in Figure 1. After 28 days post-inoculation, parasitaemia was not observed. Rabbits B1 and B2 exhibited positive bands on PCR for the persistence of Babesiosis on day 8 post-infection, as shown in Figure 2.

### 3.2. The Effect of Babesiosis on Body Temperature, Parasitaemia, and Packed Cell Volume in Experimental Calves

The calves in the infected group showed significant increases in body temperature (38.79 ± 0.03 °C) on the third day and a maximum temperature (41.3 ± 0.17 °C) on the 11th day. During the 21st and 28th day of the calves lives, two calves expired among the infected group. This study observed high fever, anaemia, and haemoglobinuria as clinical signs. A post mortem examination revealed intravascular haemolysis, splenomegaly, hepatomegaly, a dark reddish colour in the kidneys, and reddish-brown urine in the bladder. Statistically, there was no significant difference between inoculated and control group calves regarding temperature PI, as shown in Figure 3A.

All the calves in the infected group revealed parasitaemia in blood smears from the 4th day onwards. There was an increase in parasitaemia of 32%. Both the inoculated and control group calves did not exhibit parasitaemia compared to those of the infected group, as indicated in Figure 3B.

A reduction in packed cell volume (PCV) was observed in infected calves post-inoculation compared with inoculated calves. A normal PCV was recorded for cattle that had been inoculated and control group. PCV differed significantly between inoculated and infected calves as shown in Figure 3C.

## 4. Discussion

In the present study, *B. bigemina* was propagated successfully in a natural mammalian host (rabbit) with the intent of attenuation. Moreover, while several effective aspects were associated with the attenuation of *B. bigemina* in calf by passages, some serious drawbacks limited the use of this method. It is also possible that the blood of donors contains the bovine leucosis virus, which requires the adoption of strict prophylactic measures [38]. Using a rabbit model, *B. bigemina* infections of bovine red blood cells were administered intraperitoneally to induce parasitaemia. Several previous studies confirmed that circulating blood is the primary site of parasitic proliferation. A previous study examined the proliferation of *T. sergenti* in intact spleen SCID mice after the inoculation of bovine RBCs infected with *T. sergenti* intraperitoneally (IP), intravenously (IV), or subcutaneously (SC) after periodic transfusions with 1 mL of blood from healthy cattle. On day 20 after intravenous inoculation, up to 20% of mice had parasitaemia. Peak parasitaemia of 40% was recorded on the 30th day. In mice inoculated intravenously with *T. sergenti*, rapid parasitaemia development indicated that the blood was the primary proliferation site instead of the peritoneal cavity [33]. Our study determined that parasitaemia existed in rabbits for 28 days and increased to 2–4% on the fourth day post-inoculation. Similar findings were reported based on a premonition naturally developed in young animals when exposed to *B. bigemina* [39].

In this study, clinical parameters were assessed in inoculated, infected, and control group cattle calves. *B. bigemina*-infected calves showed high fever during this period, which was similar to previous studies showing fever and anaemia in cattle infected with *Babesia* [36,40]. All infected calves showed elevated body temperatures beginning on the 3rd post-infection day and reaching a maximum of 41.3 °C on the 11th post-infection day. The results of this study support those of the previous study in terms of the persistence of peak temperature increases from the 3rd to 11th days in calves carrying mixed infections of *T. annulata* and *B. bigemina* [31]. Moreover, parasitaemia increased significantly among infected calves from day 4 to day 9. The results are in line with earlier reports describing a gradual increase in parasitaemia, along with an increase in body temperature in calves that were infected with *T. annulata* [41]. In this study, the body temperature, parasitaemia, and packed cell volume (PCV) of both inoculated and control group calves were normal compared to infected calves. We also observed a slight rise in body temperature among inoculated calves without the manifestation of parasitaemia and variation in PCV. It has been reported that reduced PCV correlates with anaemia due to Babesiosis and the phagocytosis of parasitized and healthy RBCs by macrophages [42,43]. There was also a correlation between body temperature, parasitaemia, and PCV in cattle infected with B. bovis [42]. *B. bigemina*-infected calves showed decreased PCV from the 7th day onwards, with a minimum decrease on the 14th day. Similar results with reduced PCV were also reported [44] in clinically infected calves with *B. bigemina*.

## 5. Conclusions

In conclusion, this study aimed to attenuate *B. bigemina* in rabbit models used as unnatural hosts. A significant rise in parasitaemia and temperature was observed in rabbits following the inoculation of red blood cells infected with *B. bigemina*. The naive calves did not show symptoms of parasitaemia or temperature elevation when inoculated with rabbit-propagated *B. bigemina*-infected RBCs. Furthermore, this study also demonstrated that infected cattle periodically had a decreased PCV profile along with an increased temperature and parasitaemia. In the future, this study will be beneficial in the development of rabbit-attenuated *B. bigemina* isolates for further immunological studies.

## Figures and Tables

**Figure 1 animals-12-02287-f001:**
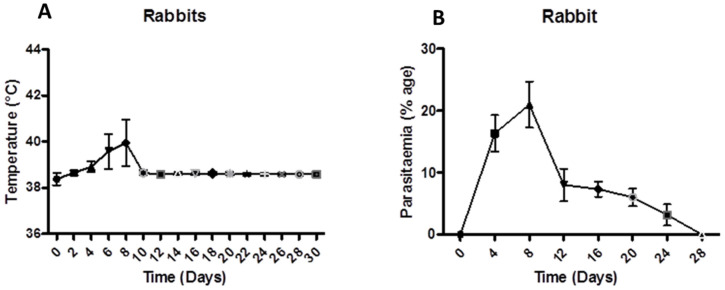
The effect of Babesiosis on body temperature (**A**) and parasitaemia (**B**) in experimental rabbits.

**Figure 2 animals-12-02287-f002:**
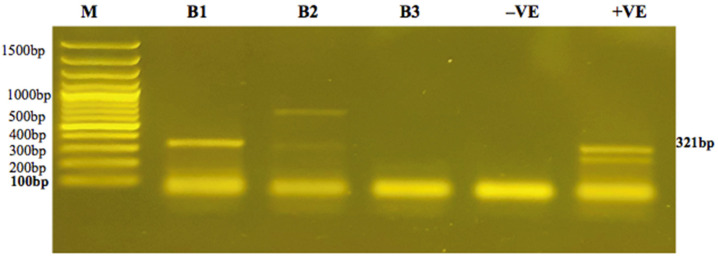
The expression of infection on PCR in experimental rabbits. Lane M: molecular weight marker of 100 bp, Cat. # SM0323; lanes B1–B3: rabbit blood samples; lane −VE: DEPC water; lane +VE: positive control of *B. bigemina,* 321 bp (Animal Disease Research Unit and Department of Veterinary Microbiology and Pathology, Washington State University).

**Figure 3 animals-12-02287-f003:**
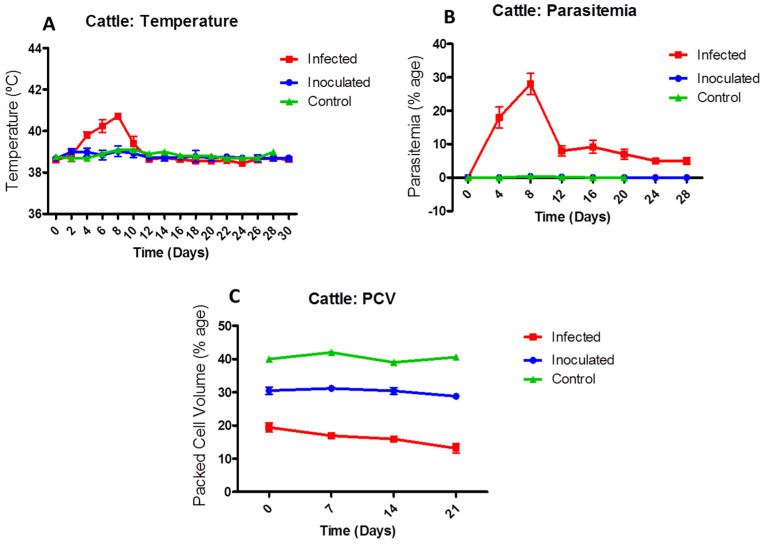
The effect of Babesiosis on body temperature (**A**), parasitaemia (**B**) and packed cell volume (**C**) in *B. bigemina*-inoculated and -infected calves.

## Data Availability

The data presented in this study are available on request from the corresponding author.

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
