# Peer review of "Propagation of Babesia bigemina in Rabbit Model and Evaluation of Its Attenuation in Cross-Bred Calves"

_animals, 2022, doi:10.3390/ani12172287_

Round 1

Reviewer 1 Report

The evaluator wants to make it clear that these comments are translated into English by means of a translator, so some terms may not be understandable. He is willing to clarify any comment made here, to the authors of the work in case what he has written is not understood.

Author Response

Reviewer’s comments-1 (Major Revision)

Manuscript ID: Animals-1839887

Dear Editor in Chief of the MDPI Journal (Animals),

Thank you very much for giving us an opportunity to revise our manuscript. We appreciate the editor and reviewers very much for their constructive comments and suggestions on our manuscript entitled

Propagation of Babesia bigemina in Rabbit model and evaluation of its attenuation in Cross-Bred Calves” (Animals-1839887).

We have studied reviewers` comments carefully. According to the reviewers’ detailed suggestions, we have made a careful revision on the original manuscript. All revised portions are marked in red in the revised manuscript which we would like to submit for your kind consideration.

Kind regards.

Prof. Dr. Kamran Ashraf

Professor at Department of Parasitology, Faculty of Veterinary Sciences, University of Veterinary and Animal Sciences, Lahore 54000, Pakistan Phone # +92321-4213897.

E-mail: kashraf@uvas.edu.pk

Corresponding author: Kamran Ashraf

https://orcid.org/0000-0002-5673-2310

Dear Editor in chief and reviewer:

Thank you for your letter and the reviewer’ comments on our manuscript entitled “Propagation of Babesia bigemina in Rabbit model and evaluation of its attenuation in Cross-Bred Calves” (Animals-1839887). Those comments are very helpful for revising and improving our paper, as well as the important guiding significance to other research. We have studied the comments carefully and made corrections which we hope meet with approval. The main corrections are in the manuscript and the responds to the reviewers’ comments are as follows.

Replies to the reviewer-1’ comments:

Question: Line 64 to 66; the referred work explains that the observation of the hemolymph of engorged females only detects a small proportion, but this does not mean that there is a low transmission of the infection to the larvae. This detection method underestimates the infectivity of the larvae. At no time does this work recommend artificial control of ticks? The direct examination of haemolymph from engorged female ticks was able to detect only a proportion of those that would transmit the parasite to their progeny. Particularly with B. argentina this method of examination underestimated considerably the infectivity of the larvae". Mahoney, D.; Mirre, G. Bovine babesiasis: estimation of infection rates in the tick vector Boophilus microplus (Canestrini). 284 Annal. Trop. Med. Parasitol. 1971. 65. 309-317.

Response: Dear Reviewer, The transmission of infection through tick larvae is irrelevant to my introduction from line no. 64-66 and so these sentences are removed as per comments of reviewer.

Question: Line 68 to 70; Publication 21 Taylor, R.; McHardy, N. Preliminary observations on the combined use of imidocarb and Babesia blood vaccine in cattle, does not speak of protection of 4 and 8 months after the use of imidocarb. The second citation, 20, Bovine babesiosis in Ireland. JS Gray, TM Murphy – Irish Veterinary News, 1985 - may explain something, but I was not able to access it as it is not online.-

My personal experience, and several articles on the subject, has always stated that much more protection is not achieved than 30 to 35 days in the best of cases, far from 4 and 8 months of protection with a single dose of imidocarb that is mentioned there?

Response: Dear reviewer, In line no. 68-70, the Publication of Taylor, R.; McHardy, N. described the adequate premunition potential of Imidocarb against both B. bovis and B. bigemina at the dose rate of 3 mg/kg as chemoprophylaxis after inoculation of bivalent Babesia vaccine on 21 and 61 days, respectively. We have also incorporated these changes in line no. 68-71. Due to unavailability of citation, reference 20, has also been removed from these lines.

Question: The materials and methods are not clear or ordered, In source of parasites, lines 107 to 121 it explains that healthy rabbits were purchased for the experiment, which should go in 2.3 inoculation into rabbit, since there it does not explain how many rabbits were inoculated, and one concludes that they will be 6 because they read it above. Line 125, protocol used somewhere [35] (It does not seem correct to speak of somewhere here, when a work carried out on mice is cited that can possibly be extrapolated.)?

Response: Dear reviewer, we have incorporated the lines 107 to 121 in the 2.2 section of our Materials and Methods. The changes have been incorporated as per comments of the reviewer. Six male and healthy rabbits were inoculated with 3x106 infected RBCs with B. bigemina intraperitoneally simultaneously. Line 122, has also been updated with description of the protocol used earlier instead of elsewhere.

Question: 2.4 Line 136 to 140; Healthy calves are inoculated with parasitized blood from rabbits, but it does not explain how many rabbits this blood was obtained from, on what days it was obtained and what Parasitaemia this blood had when the calves were inoculated, to reach a dose of 2x106 parasitized erythrocytes. Then they bought 10 calves, (we need to explain the weight and sex of these animals) and 5 healthy ones were inoculated with rabbit blood with Babesia bigemina and 5 infected said to be used as control, but it is not clear what infected calves mean. (are they naturally infected?)?

Response: There were six rabbits used for obtaining parasitized blood with 4% Parasitaemia and calves in inoculated group were inoculated with a dose of 2x106 parasitized erythrocytes subcutaneously on 7th day post-propagation of B. bigemina. There were total fifteen (n=15) 4-8 months old, male calves weighing about 60-80 kg were used for this study instead of ten. They were also screened for any infection through microscopy and PCR. Based on screening, these calves were divided into three groups: inoculated, infected and control groups. Each group contained (n=5) calves, respectively. Infected group contained experimentally infected calves following the protocol adopted earlier (Rauf et al, 2020). We have incorporated all these changes in 2.3 section of Materials and Methods in line no. 129-131 and 134-139 as per comments of the reviewer.

Question: "Healthy animals were inoculated with 2x106 B. of bigemina-infected RBCs subcutaneously, whereas infected animals were used as controls." The definition of infected and inoculated is not clear, it never speaks of a negative control, and it is not shown on the graph. I must suspect that a negative control (calves inoculated with un-parasitized rabbit blood) was not used for this experiment?

Response: These healthy animals (n=5) were cattle calves from inoculated group that were inoculated with 2x106 B. bigemina infected RBCs subcutaneously. The infected group contained five experimentally infected calves post-screening. While another group containing five healthy calves served as negative control group in this study for comparison of Temperature, Parasitaemia and Packed Cell Volume (PCV). We have incorporated these changes in line no. 135-138 of 2.3 sections of Materials and Methods as per comments of the reviewer.

Question: 3. Line 168 - 169 He mentions that the temperature in the infected calves increased in a range from 38.79 +-0.03 to 38.98 +-0.17, but the graph shows a temperature of 41 ºC on those days.

He continues mentioning infected and inoculated, but the evaluator does not understand the term infected?

Response: In Line no. 168 – 169, the temperature in the infected calves increased in a range from 38.79 +-0.03 to 41.3 +-0.17 instead of 38.98+-0.17,  as per graph on those days. We have incorporated the change of temperature in line no. 170-172 of 3.2 sections of our Results as per recommendation of reviewer comments.

Question: Line 181 A higher level of protection was also observed among all the vaccinated calves as shown in Figure 3C.

The graph quoted shows the PVC, not a level of protection against Babesia bigemina.

Response: In Line no. 181, a higher level of protection was also observed among all the vaccinated calves instead of inoculated calves is irrelevant to my study. Therefore this sentence is been removed from this line as per comment of the reviewer.

Question: Line 209 “All infected calves showed elevated body temperatures beginning on the third post-infection day and reaching a maximum of 41.3 °C on the 11th post-infection day”.

The above results do not speak of that temperature (line 181)?

Response: Line 209 “All infected calves showed elevated body temperatures beginning on the third post-infection day and reaching a maximum of 41.3 °C on the 11th post-infection day” has been described in Results section 3.2 in the line no. 170-172 as per comments of the reviewer.

Question: Conclusions should be better. It remains unclear whether calves inoculated with the blood of infected rabbits produced antibodies against Babesia bigemina, part of the initial objective of this work was the possibility of developing a vaccine in the future. "Therefore, the aim of the present study was to attenuate B. bigemina in rabbits for the purpose of developing a vaccine against Babesiosis. We hypothesized that B. bigemina can be propagated in vivo in rabbits, and that it can be attenuated in cross- bred calves." (Line 88-92)?

Response: In the Line no. 88-92, aim of the study based on hypothesis “propagation of B. bigemina in rabbit model and evaluation of its attenuation in naïve calves” based on clinical parameters i.e., temperature, parasitaemia and packed cell volume (PCV) and has been updated. There has no study conducted based on evaluation of evaluation of antibodies production against Babesia bigemina in inoculated calves with in vivo attenuated B. bigemina in rabbit model. We have incorporated the changes as per comments of the reviewer.

Question: Bibliography 32 and 34 is not referenced at the work.-?

Response: 32 and 34 references have been incorporated in the line no. 107 and 113.

Reviewer 2 Report

Reviewer report:

The manuscript by Ullah et al., present results and data on Propagation of Babesia bigemina in Rabbit model and evaluation of its attenuation in Cross-Bred Calves.

The authors analyzed propagation of Babesia bigemina in vivo in rabbit for attenuation and evaluated its virulence periodically and inoculated attenuated B. bigemina to naive calves for evaluation of clinical parameters.

the information is interesting and important for researchers in the field of animal diseases and bovine babesiosis.

The study needs some improvement in both methodology and results sections to improve the manuscript.

Materials and methods

Line 115: please state the gene name for the B. bigemina primers and a reference to the PCR- primers and protocol. It also would be great if there is a sequence confirming the origin of the PCR product of B. bigemina

Line 123: It is not clear if all the rabbits are inoculated at the same time or was it one after the other like successive passage?

Line 137: Infected animals were used as a control, It is concerning that from 10 calves five were healthy and other five were sick with B. bigemina and B. bigemina only? Is there a 50 % of the cattle in the market are sick? And with B. bigemina only? Please clarify.

Line 174: Inoculated and Un-inoculated calves, the authors mentioned infected and inoculated calves many times please keep one of them through the paper.   

Results

Line 182: What do you mean by vaccinated calves.

All figures look clear,

Discussion

Regarding infected group, it was not clear if it is experimentally infected calves or sick animals that were naturally infected, Using naturally infected calves as a control is risky, if there were any naturally infected cattle the peak of parasitemia and disease symptoms are different from experimentally infected animals because sometimes the naturally infected animals has chronic infection with low PPE.

Author Response

Reviewer’s comments-2 (Major Revision)

Manuscript ID: Animals-1839887

Dear Editor in Chief of the MDPI Journal (Animals),

Thank you very much for giving us an opportunity to revise our manuscript. We appreciate the editor and reviewers very much for their constructive comments and suggestions on our manuscript entitled

Propagation of Babesia bigemina in Rabbit model and evaluation of its attenuation in Cross-Bred Calves” (Animals-1839887).

We have studied reviewers` comments carefully. According to the reviewers’ detailed suggestions, we have made a careful revision on the original manuscript. All revised portions are marked in red in the revised manuscript which we would like to submit for your kind consideration.

Kind regards.

Prof. Dr. Kamran Ashraf

Professor at Department of Parasitology, Faculty of Veterinary Sciences, University of Veterinary and Animal Sciences, Lahore 54000, Pakistan Phone # +92321-4213897.

E-mail: kashraf@uvas.edu.pk

Corresponding author: Kamran Ashraf

https://orcid.org/0000-0002-5673-2310

Dear Editor in chief and reviewer:

Thank you for your letter and the reviewer’ comments on our manuscript entitled “Propagation of Babesia bigemina in Rabbit model and evaluation of its attenuation in Cross-Bred Calves” (Animals-1839887). Those comments are very helpful for revising and improving our paper, as well as the important guiding significance to other research. We have studied the comments carefully and made corrections which we hope meet with approval. The main corrections are in the manuscript and the responds to the reviewers’ comments are as follows.

Replies to the reviewer-2’ comments:

Question: Materials and methods; Line 115: please state the gene name for the B. bigemina primers and a reference to the PCR- primers and protocol. It also would be great if there is a sequence confirming the origin of the PCR product of B. bigemina?

Response: The gene was 18srRNA gene for specific primers of B. bigemina used while performing PCR. It has also been used in previous study (Zaheer et al., 2020) that is also cited in protocol. There is no confirmed sequence regarding to the origin of the PCR product of B. bigemina but we have confirmed through comparison with control positive of B. bigemina available to us for this study. According to the reviewers’ detailed suggestions, we have made a careful revision in the line no.113.

Question: Line 123: It is not clear if all the rabbits are inoculated at the same time or was it one after the other like successive passage?

Response: All the rabbits were inoculated simultaneously and were periodically evaluated for temperature and Parasitaemia. According to the reviewers’ detailed suggestions, we have incorporated the changes in line no. 122.

Question: Line 137: Infected animals were used as a control, it is concerning that from 10 calves five were healthy and other five were sick with B. bigemina and B. bigemina only? Is there a 50 % of the cattle in the market are sick? And with B. bigemina only? Please clarify.

Response: These were actually total fifteen (n=15) calves that were screened for any infection through microscopy and PCR. Based on screening, these calves were divided into three groups: inoculated, infected and control groups. Each group contained (n=5) calves, respectively. Among the healthy calves, half calves were inoculated with 2x106 B. bigemina-infected RBCs subcutaneously and served as Inoculated group, whereas (n=5) healthy animals served as negative control. Infected group contained five calves experimentally infected following the protocol adopted earlier (Rauf et al., 2020).These are written as groups: healthy and infected calves instead of inoculated, infected and control groups. According to the reviewers’ detailed suggestions, we have made changes in the line no. 134-139.

Question: Line 174: Inoculated and Un-inoculated calves, the authors mentioned infected and inoculated calves many times please keep one of them through the paper.  

Response: These are actually inoculated and negative control calves instead of Un-inoculated ones. According to the reviewers’ detailed suggestions, we have incorporated the changes in the line no. 176-177.

Question: Results; Line 182: What do you mean by vaccinated calves?

Response: Dear reviewer, these are actually inoculated calves instead of vaccinated ones and are irrelevant to my study. Therefore this sentence is been removed from this line as per comment of the reviewer.

Question: Discussion; Regarding infected group, it was not clear if it is experimentally infected calves or sick animals that were naturally infected, Using naturally infected calves as a control is risky, if there were any naturally infected cattle the peak of Parasitaemia and disease symptoms are different from experimentally infected animals because sometimes the naturally infected animals has chronic infection with low PPE.

Response: Infected cattle group involved in this research study was experimentally infected and it has also been mentioned in methodology by following the protocol adopted earlier (Rauf et al., 2020). We have incorporated the changes in line no. 134-139 as per comments of the reviewer.

Round 2

Reviewer 1 Report

The evaluator wants to make it clear that these comments are translated into English by means of a translator, so some terms may not be understandable. He is willing to clarify any comment made here, to the authors of the work in case what he has written is not understood.

The review considers that the work has improved a lot in terms of the explanation of materials, methods and results. There are still some small details that could be improved in my opinion. Anyway I put here some considerations still seen.

line 135. "whereas rest of healthy calves (n=5) served as negative control group"
The reviewer considers that to be a negative control group, these animals should be inoculated with blood from healthy rabbits, without Babesia bigemina. Only to demonstrate that the signs and symptoms found in the inoculated animals are due to the presence of Babesia bigemina, and not to some other component of rabbit blood that has given a reaction in the inoculated calves, and results from those signs. and symptoms explained at work. This is not detailed in the work, it only says that 5 healthy animals are used as a negative control. It is also strange that they have forgotten to put this group in the original work, considering that animals are bought (initially 10 are bought, but later 15 were bought).

line 232 "In the future, this study may contribute to the development of a vaccine against B. bigemina. "

Even with these preliminary studies, and without a determination of the level of antibodies in the calves, and then a challenge with Babesia bigemina  field strain (pathogenic) to these animals by the inoculation of the blood of rabbits, nothing indicates to me of this work,  that this methodology can be used for a subsequent vaccination against Babesia bigemina.

There are still many studies to reach this conclusion, and I would not put it as part of the conclusions.

Author Response

Reviewer’s comments (Minor Revision)

Manuscript ID: Animals-1839887

Dear Editor in Chief of the MDPI Journal (Animals),

Thank you very much for giving us an opportunity to revise our manuscript. We appreciate the editor and reviewers very much for their constructive comments and suggestions on our manuscript entitled

Propagation of Babesia bigemina in Rabbit model and evaluation of its attenuation in Cross-Bred Calves” (Animals-1839887).

We have studied reviewers` comments carefully. According to the reviewers’ detailed suggestions, we have made a careful revision on the original manuscript. All revised portions are marked in red in the revised manuscript which we would like to submit for your kind consideration.

Kind regards.

Prof. Dr. Kamran Ashraf

Professor at Department of Parasitology, Faculty of Veterinary Sciences, University of Veterinary and Animal Sciences, Lahore 54000, Pakistan Phone # +92321-4213897.

E-mail: kashraf@uvas.edu.pk

Corresponding author: Kamran Ashraf

https://orcid.org/0000-0002-5673-2310

Dear Editor in chief and reviewer:

Thank you for your letter and the reviewer’ comments on our manuscript entitled “Propagation of Babesia bigemina in Rabbit model and evaluation of its attenuation in Cross-Bred Calves” (Animals-1839887). Those comments are very helpful for revising and improving our paper, as well as the important guiding significance to other research. We have studied the comments carefully and made corrections which we hope meet with approval. The main corrections are in the manuscript and the responds to the reviewers’ comments are as follows.

Replies to the reviewer’s comments:

Question: line 135. "whereas rest of healthy calves (n=5) served as negative control group"
The reviewer considers that to be a negative control group, these animals should be inoculated with blood from healthy rabbits, without Babesia bigemina. Only to demonstrate that the signs and symptoms found in the inoculated animals are due to the presence of Babesia bigemina, and not to some other component of rabbit blood that has given a reaction in the inoculated calves, and results from those signs. and symptoms explained at work. This is not detailed in the work, it only says that 5 healthy animals are used as a negative control. It is also strange that they have forgotten to put this group in the original work, considering that animals are bought (initially 10 are bought, but later 15 were bought)?

Response: In the first draft we did not include the control/uninfected group as a similar publication from our research group published in Animals (Ramzan et al., 2022) (cited: Animals 2022, 12, 813. https://doi.org/10.3390/ani12070813).We based that study on comparison of attenuated group vs non-attenuated group for analysis and the reviewers agreed to it for publication.

Now, we have added the data from control/uninfected group, which was inoculated with similar dose of un-infected RBCs from healthy rabbit. We also have incorporated these changes in line no. 135 as per valuable comments of the reviewer.

Moreover, we followed the control/uninfected group in parallel with other groups, as shown in the Fig. 3, we monitored the temperature, parasitaemia and PCV of these calves.  

It is pertinent to mention here that we did not inoculated whole blood from rabbits to the calves, only the packed cell volume having counted RBCs were used for inoculation. Therefore, there are no chances of any reaction to some other component of rabbit blood in calves as mentioned by the reviewer.

Question: line 232 "In the future, this study may contribute to the development of a vaccine against B. bigemina. “Even with these preliminary studies, and without a determination of the level of antibodies in the calves, and then a challenge with Babesia bigemina field strain (pathogenic) to these animals by the inoculation of the blood of rabbits, nothing indicates to me of this work,  that this methodology can be used for a subsequent vaccination against Babesia bigemina. There are still many studies to reach this conclusion, and I would not put it as part of the conclusions.

Response: We did not check any kind of immune response so, it is not the vaccine study. The purpose of our study was to check virulence of Babesia bigemina after passaging in rabbit. In the future, this study will be beneficial in the development of rabbit-attenuated B. bigemina isolates for further immunological studies. We have incorporated these changes in line no. 232 as per comments of the reviewer
